# Matcha Does Not Affect Electroencephalography during Sleep but May Enhance Mental Well-Being: A Randomized Placebo-Controlled Clinical Trial

**DOI:** 10.3390/nu16172907

**Published:** 2024-08-31

**Authors:** Yoshitake Baba, Takanobu Takihara, Noritaka Okamura

**Affiliations:** 1Central Research Institute, ITO EN, Ltd., 21 Mekami, Makinohara 421-0516, Shizuoka, Japan; t-takihara@itoen.co.jp; 2Department of Medical Technology, Faculty of Health Sciences, Ehime Prefectural University of Health Sciences, 543 Takoda, Tobe-cho 791-2101, Iyo-Gun, Japan; okamu@epu.ac.jp

**Keywords:** matcha, theanine, caffeine, EEG, sleep quality, depression

## Abstract

Although theanine in matcha improves sleep quality and cognitive function, the caffeine in green tea is thought to worsen sleep quality. Therefore, this study investigated the factors behind the observed improvements in subjective sleep quality in matcha. A placebo-controlled randomized double-blind parallel-group study was conducted on healthy Japanese men and women aged 27–64 years. After 4 weeks of consuming 2.7 g of matcha daily (containing 50.3 mg theanine, 301.4 mg catechins, and 71.5 mg caffeine), no significant differences were observed between the control and matcha groups on total sleep time, sleep latency, wake after sleep onset, or sleep efficiency measured by electroencephalography (EEG). However, the sleep questionnaire Oguri–Shirakawa–Azumi Sleep Inventory, the Middle-age and Aged version (OSA-MA), administered immediately after waking showed a trend toward increased satisfaction with sleep time (*p* < 0.1), and EEG measurements indicated significantly shortened wake-up times after waking with matcha intake (*p* < 0.05). The Beck Depression Inventory-II scores also tended to decrease (*p* < 0.1). The continuous intake of matcha may offer improved subjective sleep quality and emotional stability despite not offering significant changes in objective sleep parameters.

## 1. Introduction

The health benefits of matcha have attracted attention in recent years. Matcha is a green powder made by finely grinding green tea leaves. The raw material for matcha is called tencha, from which the veins and stems have been removed. One of these benefits is the improvement in cognitive function. Matcha improves attention and working memory through the effects of theanine [1], caffeine [2], and catechins [3]. In an intervention study aimed at improving cognitive function in individuals with a diagnosis of subjective cognitive decline and mild cognitive impairment, continuous intake of matcha for 1 year resulted in improved facial expression recognition, social cognitive function (*p* < 0.05), and a tendency toward improved subjective sleep quality, assessed by the Japanese version of the Pittsburgh Sleep Quality Index (PSQI-J) (*p* < 0.1) [4]. As of 2023, Japan is a super-aging society with an aging rate of 29.1%, among the highest in the world [5], raising concerns regarding an increase in patients with dementia. Dementia is characterized by cognitive decline, and its early stages often include a deterioration in sleep quality, which is thought to increase the risk of dementia [6]. Green tea [7] and matcha [2,8,9]—traditional Japanese beverages—have been shown to prevent cognitive decline.

Lack of sleep is a potential health problem in Japan. According to the National Health and Nutrition Survey conducted by the Ministry of Health, Labor, and Welfare in 2019, 34.6% of people sleep between 6–7 h, followed by 30.3% who sleep between 5–6 h [10]. However, the sleep time of Japanese individuals is shorter than that of other countries [11,12]. This raises concerns regarding a lack of attention [13,14]. Therefore, exploring ways to improve sleep quality is an important issue to enhance quality of life. There have been no reported cases regarding the effect of matcha, a type of green tea, on sleep quality. Since the caffeine contained in coffee worsens sleep quality [15], it is thought that green tea and matcha, which also contain caffeine, also worsen sleep quality. Reports have stated that green tea with reduced caffeine improves sleep quality [16]. Therefore, in Japan, it is customary to avoid consuming green tea before going to bed. In a previous study in which participants consumed matcha continuously for 1 year [4], they were also asked to refrain from consuming the test food before going to bed to prevent a deterioration in sleep quality.

Dementia is characterized by a decline in cognitive functions such as memory, executive function, attention, language, social cognition and judgment, psychomotor speed, and visual or visuospatial cognition, compared to previous levels [17]. Sleep is important for brain rest and recovery. It is believed that memory consolidation, which involves transferring memories formed in the hippocampus to the cerebral cortex, occurs during sleep [18]. Thus, deterioration in sleep quality may contribute to cognitive decline.

Theanine is a non-protein amino acid that may contribute to improved cognitive function and sleep quality. It is found in tea trees (*Camellia sinensis* L.) [19] and its content increases when grown in the shade for a certain time [20]. Theanine intake changes the concentration of neurotransmitters, such as dopamine, serotonin [21], tryptophan, and gamma-aminobutyric acid (GABA) [22]. GABA is an inhibitory neurotransmitter. Furthermore, theanine inhibits glutamine transport in neurons and astroglia in the rat brain [23], alleviating the excitotoxicity of glutamate. These changes in neurotransmitters lead to beneficial effects such as anti-stress effects [24] and sleep improvement effects. The intake of 200 mg of theanine produced sleep improvement effects and enhanced objective sleep quality, which was assessed using the Oguri–Shirakawa–Azumi Sleep Inventory, the Middle-age and Aged version (OSA-MA), sleep questionnaire [25]. Conversely, studies have reported that caffeine inhibits sleep when consumed at a dosage of ≥100 mg. The amount of caffeine contained in green tea is 0.02 g/100 g (sencha infusion) [26], and in the 2070 mg of matcha used in a previous intervention study [4], it was 66.2 mg. This was less than the amount that is believed to affect sleep. In contrast, theanine was found to suppress caffeine-induced awakenings in a study evaluating the effect of simultaneous administration of theanine and caffeine on sleep quality [27]. Participants ingested 30 mg of caffeine and 50 mg of theanine and underwent electroencephalography (EEG) measurement [28]. The results showed that theanine suppressed a significant increase in awakening time induced by caffeine.

Thus, theanine and caffeine have opposite effects on neural activity. Caffeine [29] acts as a stimulant, whereas L-theanine [30] acts as a neural depressant. The tea tree contains both of these compounds, which may interact beneficially in some cases [31], such as enhancing attention during wakefulness. However, it is improbable that matcha, which contains caffeine in quantities that do not affect sleep, and theanine, in amounts lower than those known to improve sleep quality, would enhance sleep. Therefore, this study aimed to evaluate sleep quality using EEG and sleep quality surveys to explore the factors contributing to the potential sleep-enhancing effects of matcha. In addition, participants’ levels of depression (measured by Beck’s depression index [BDI-II]) and psychological factors were surveyed.

## 2. Materials and Methods

### 2.1. Ethics Statement

The protocol of this placebo-controlled parallel group study was approved by the Ethics Review Committee of Watanabe Hospital (Tokyo, Japan; approval number: SL1-2305). The study was conducted in accordance with the tenets of the Declaration of Helsinki (adopted in 1964 and amended in 2013) and was carried out at Watanabe Hospital (Tokyo, Japan) between September 2023 and December 2023. It was registered at the University Hospital Medical Information Network (UMIN; Tokyo, Japan; number, UMIN 000052248).

### 2.2. Test Food

The matcha used in this study was sourced from ITO EN, Ltd. (Tokyo, Japan), and was the first flush from Kagoshima Prefecture. The total contents of theanine, catechins, and caffeine in the daily intake of matcha (2700 mg) were 50.3 mg, 301.5 mg (epigallocatechin gallate 143 mg, gallocatechin gallate 0.8 mg, epicatechin gallate 24.6 mg, catechin gallate 0.1 mg, epigallocatechin 106 mg, gallocatechin [GC] 4.5 mg, epicatechin [EC] 20.9 mg, and catechin [C] 1.6 mg), and 71.5 mg. The participants took 15 capsules per day. White, porcine, gelatin, and number 1 capsules (the Japanese Pharmacopoeia) were used for both the placebo and matcha capsules. The placebo capsules were green and filled with starch. 

### 2.3. Participants

The participants recruited for the study were healthy Japanese men and women aged ≥ 20 years. The eligibility criteria included (1) individuals experiencing difficulty sleeping, (2) those capable of maintaining an appropriate bedroom temperature for sleep, and (3) women with a stable menstrual cycle. The exclusion criteria were individuals (1) who were students; (2) currently undergoing treatment for a severe illness; (3) working night shifts, rotating shifts, or engaged in heavy labor; (4) with irregular weekday sleep hours or habits; (5) currently receiving treatment for mental disorders (such as depression) or sleep disorders or with a history of such disorders; (6) diagnosed with sleep apnea syndrome or aware of having apnea; (7) experiencing nocturnal urination symptoms or undergoing treatment for urinary system disorders; (8) individuals selecting “I want to commit suicide” or “If given the opportunity, I would commit suicide” on question 9 of the BDI-II; (9) who were smokers; (10) consuming excessive amounts of alcohol or caffeine (diagnosed with alcoholism or caffeine addiction); (11) rarely consuming caffeinated beverages such as green tea or coffee; (12) habitually consuming foods that inhibit sleepiness before bedtime or using devices that interfere with sleepiness; (13) with metal allergies (causing skin problems from electrodes) or sensitive skin; (14) caring for preschool children or others, sleeping with two or more individuals in the same bed or futon, or potentially affected by others causing sleep disturbances; (15) with food allergies or those susceptible to allergies related to test foods; (16) regularly using medications or quasi-drugs that may affect the study outcomes; (17) regularly using health foods (such as specified health use foods, nutritional function foods, foods with functional claims, supplements) that may affect the study outcomes; (18) currently enrolled in a clinical trial or having participated in a clinical trial within the past month from the consent date; (19) unable to maintain their usual daily lifestyle during the study period; (20) whose bedtime schedule might have changed during the study period owing to overseas travel or business trips; (21) who were pregnant, breastfeeding, or intending to become pregnant; (22) with severe symptoms of premenstrual syndrome or menopausal disorders; and (23) planning to participate in another study during the study period.

### 2.4. Study Design

A placebo-controlled randomized double-blind parallel-group study was conducted. The flow diagram is shown in Figure 1. The primary endpoint was the index obtained from EEG, total sleep time, sleep latency, wake after sleep onset, number of awakenings after sleep onset, sleep efficiency, and time spent in bed after waking. The secondary endpoint was the results of the PSQI-J, OSA-MA, Japanese version of the Epworth Sleepiness Scale (JESS), and BDI-II. 

In a previous randomized placebo-controlled crossover comparative study [25] that evaluated the effects of L-theanine intake on sleep indices, the mean ± standard deviation time to wakefulness after sleep onset was 19.8 ± 7.6 in the placebo intake condition and 12.6 ± 4.5 in the L-theanine intake condition. Based on these results, the effect size of L-theanine in healthy young males was estimated to be approximately 1.2 ((19.8–12.6)/6.1). Similarly, sleep efficiency was 93.8 ± 3.0% in the placebo intake condition and 96.6 ± 1.3% in the L-theanine intake condition, with an estimated effect size of approximately 1.3 [25]. Unlike the crossover study, the main analysis of the current study was the comparison of unpaired data (assuming a *t*-test), because it targeted young men and men and women across a wider age range. Therefore, variability in the intervention effect among participants was expected to be greater than that in the reference study. However, even with a smaller effect size of approximately 0.8, which is below the two-sided significance level of 5%, a sample size of 60 cases (30 cases per group) would have provided adequate statistical power (≥80%). After pre-intervention measurements, conducted in late September 2023, the participants were assigned to either a placebo or matcha group, and post-intervention measurements were conducted 4 weeks after ingesting the test food in early December 2023. The participants were instructed not to consume the test food within 2 h before bed because it contains caffeine. 

### 2.5. Participant Evaluations

An EEG was measured using a wearable polysomnography device to evaluate objective sleep quality (Insomnograf M2; S’UIMIN Inc., Tokyo, Japan). The device consists of a main body and electrodes and can measure brainwaves during sleep with the same accuracy as a polysomnograph. The agreement rate with a general polysomnography device was 86.9% and the kappa coefficient was 0.8015 [32]. The measurements were conducted at participants’ homes, ensuring the indoor environment matched their usual sleep conditions. Five electrodes were attached according to the 10/20 method, including Fp1, Fp2, A1, A2, and the area between Fp1 and Fp2 as Ref. The electrodes for Fp1, Fp2, and Ref were integrated as a triple electrode and attached to the forehead. The electrodes were affixed with adhesive tape, and the participants applied them to the designated areas themselves. The participants were instructed on the attachment sites and how to operate the equipment before data recording. When the participants went to bed, they lay down and activated the device themselves to begin recording when they intended to fall asleep naturally. Upon waking up and getting out of bed, they turned off the device, thereby concluding the measurement.

The participants measured EEG data for 5 consecutive days, excluding Saturdays and Sundays, and the data from Tuesday, Wednesday, and Thursday were used. The following criteria were adopted when utilizing the data. (1) If lights-off and -on times for the 3 days of Tuesday, Wednesday, and Thursday were within ±1 h (60–70 min), the participant was considered as having regular sleep. (2) If the lights-off and -on times for either Tuesday, Wednesday, or Thursday were >1 h apart, the data from Monday or Friday were used, judging that the weekday sleep habits were irregular. (3) If the conditions described in 2.2 Exclusion Criteria were not met for the three usual weekdays, the data from the other two weekdays were used. The participants wore the device and went to sleep, and the following morning, they answered the OSA-MA at home to perform a subjective evaluation of the quality of their previous night’s sleep. The PSQI-J, JESS, and BDI-II were measured at the visits at 0 and 12 weeks.

Data extraction and analysis were performed by S’UIMIN Co., Ltd (Tokyo, Japan). The sleep stages were divided into wakefulness and rapid eye movement (REM), non-REM 1 (N1 [light]), non-REM 2 (N2 [medium]), and non-REM 3 (N3 [deep]) sleep. The total sleep time, number of awakenings, and duration of awakenings were also calculated.

### 2.6. Statistical Analysis

The values are reported as mean ± SD. Statistical analysis was performed using SAS version 9.4 for Windows. Before the analysis, normality was assessed. If normality was confirmed, comparisons within groups were performed using one-sample *t*-tests, and comparisons between groups were performed using two-sample *t*-tests. If normality was not confirmed, the Wilcoxon signed-rank test or Mann–Whitney U test was used. For ordinal data, the Wilcoxon rank-sum test was employed for comparisons between groups, and the Wilcoxon signed-rank test was used for comparisons within groups. The significance level was set at *p* < 0.05.

## 3. Results

The demographic characteristics of the participants analyzed are shown in Table 1. Allocation was based on age, sex, PSQI score, and sleep efficiency. The PSQI-J cutoff value is 5/6, with a score of 6 or more, is considered indicative of sleep abnormality. The participants in this study scored approximately 4 points, indicating normal sleep quality. There is no standard for determining whether sleep efficiency measured by EEG is normal or indicative of a sleep disorder. The participants’ sleep efficiency was approximately 90%.

### 3.1. Sleep Quality Measured by EEG (Total Sleep Time, Sleep Latency, Wake after Sleep Onset Time, Number of Wake after Sleep Onset, and Sleep Efficiency)

No significant differences were observed between the placebo and matcha groups in terms of total sleep time, sleep latency, duration and number of wake after sleep onsets, and sleep efficiency calculated from the EEG. Time spent in bed after waking indicated significantly lower values in the matcha group compared to the placebo group (Table 2).

### 3.2. Evaluation of Sleep Quality Using a Sleep Questionnaire

In the PSQI-J, which assesses the quality of sleep over the past month, the placebo group had significantly lower values at 4 weeks compared to 0 weeks (*p* < 0.05). The matcha group also showed similarly low values, but no significant difference was observed. The JESS, which measures daytime sleepiness, showed significantly lower values at 4 weeks compared to 0 weeks in both the placebo and matcha groups (*p* < 0.05). In the OSA-MA, which evaluates the quality of last night’s sleep, both the placebo and matcha groups showed significantly lower values for sleepiness on rising and feeling refreshed on rising at 4 weeks compared to 0 weeks (*p* < 0.05 or *p* < 0.01). Regarding frequent dreaming, the matcha group tended to show lower values compared to the placebo group (*p* < 0.1). The initiation and maintenance of sleep scores were significantly lower in the placebo group (*p* < 0.05), whereas sleep length was significantly lower in the matcha group (*p* < 0.01) at 4 weeks compared to 0 weeks. For sleep length, the matcha group tended to show a smaller change from week 0 compared to the placebo group (*p* < 0.1). For the BDI-II, used in screening for depression, the matcha group showed significantly lower values at week 4 compared to week 0 (*p* < 0.05), and the rate of change from week 0 tended to be lower compared to the placebo group (*p* < 0.1) (Table 3).

## 4. Discussion

A previous study reported that older individuals with an average age of 72 years were able to improve their facial expression recognition function by regularly consuming 2 g of matcha tea (containing 48.1 mg of theanine) for 1 year (*p* < 0.05). Additionally, as a secondary evaluation of sleep quality, we observed a tendency toward improvement in subjective sleep quality (*p* < 0.1) [4]. It is believed that sleep quality deterioration is an early stage of dementia and that it increases the risk of dementia [6]. Consequently, we aimed to verify whether matcha improves sleep quality based on objective EEG data. Matcha contains theanine, which contributes to improved sleep, and 50.3 mg of it—the same amount as in the previous study—was employed in our test.

Tea is a popular beverage worldwide [33]. Green tea contains two components that affect the brain: theanine [34] and caffeine [35]. Caffeine has a stimulating effect, whereas theanine has a calming effect. Theanine also suppresses the excitotoxicity of caffeine [30]. Caffeine blocks adenosine A2A receptors, which regulate the release of the excitatory neurotransmitter glutamate. When adenosine binds to the receptor, the release of glutamate is inhibited, suppressing excitation. However, when caffeine inhibits adenosine binding to its receptor, nerve cells become excited [36]. Other mechanisms of action for this excitatory effect include the inhibition of phosphodiesterase activity by high concentrations of caffeine [35] and indirect stimulation of dopamine D2 receptors through the blockage of adenosine receptors, especially A2A receptors [36], and is thought to affect several psychological functions [35]. We previously demonstrated that consuming matcha tea, which contains both theanine and caffeine, improves attention function [2,8]. This effect has also been observed with the ingestion of caffeine alone [2] or theanine alone [1]. 

Regarding sleep, it suppressed the increase in nighttime wakefulness caused by caffeine. Theanine has also been shown to potentially reduce the deterioration of sleep quality caused by caffeine [27]. The study involved nine healthy Japanese women aged 21–22 years and a crossover study with four groups: placebo, 50 mg theanine group, 30 mg caffeine group, and a mixed group of theanine and caffeine (Mix). The beverage was consumed immediately before going to bed. EEG measurements showed a significant increase in the time spent awake during the night in the caffeine group, whereas the theanine and Mix groups showed no significant difference compared with the placebo group. However, in this study, no increase in nighttime wakefulness was observed in the matcha group despite the higher caffeine content than that of theanine. The same was true for total sleep time, sleep latency, wake after sleep onset time, number of awakenings after sleep onset, or sleep efficiency. It is important to consider that the suppression of increased wakefulness after sleep onset, despite the higher caffeine content in this study, may be due to participants being instructed not to consume matcha within 2 h before going to bed.

In the subjective assessment of sleep quality, a decrease in PSQI-J score was observed in both the placebo and matcha groups. The change in participants’ awareness after the 1-month intervention trial itself resulted in improved sleep quality. The study also discovered that sleep is affected by environmental factors, but the measurements before the intervention were taken in September and after the intervention in December, with temperatures of 25.3 °C and 10.8 °C, respectively. Therefore, the improvement in scores can also be attributed to seasonal comfort. The OSA-MA, developed in Japan, investigates the previous night’s sleep state upon waking up [37]. OSA-MA scores showed a tendency for satisfaction with sleep time to increase with the intake of matcha (*p* < 0.1). In a study of adult males with an average age of 27.5 ± 0.9 years, 200 mg of theanine was taken for 6 days, and OSA-MA was assessed, which reported a significant improvement in sleep time [25]. In addition, the results of the actigraphy evaluation from the same participants showed improvement in sleep efficiency and reduction in wake after sleep onset. Although no significant difference was observed in the sleep time category of the OSA-MA in this study (*p* < 0.1), it does not rule out the possibility that theanine improves overall sleep quality. Frequent dreaming, as measured by the OSA-MA, is a stress-related item. Stress is one of the factors that affect sleep, and the observed tendency for improvement in this indicator (*p* < 0.1) is considered an important marker of mental health. We believe that this tendency for improvement is linked to the feeling of having slept well, as indicated by sleep length in the OSA-MA and time spent in bed after waking. A long time spent in bed after waking may indicate a state of stress. Since this time was shortened, we believe that stress was reduced.

Concerning subjective sleep quality, wake-up time calculated using EEG was significantly shorter with the intake of matcha tea. In cases of depression, individuals often experience difficulties falling asleep, waking up in the middle of the night, and waking up early in the morning. Additionally, depression is associated with a lack of deep sleep, a feeling of unrest, and difficulty getting out of bed in the morning. Therefore, difficulty getting out of bed in the morning may be influenced by psychological factors. Theanine in green tea reportedly improves symptoms of stress and depression [38]. Therefore, it can be inferred that a positive effect of theanine on mental health was also observed in the study with the BDI-II scores, showing a tendency for improvement (*p* < 0.1).

However, a limitation of this study is that the effect was observed among Japanese people who have a habit of drinking green tea. Therefore, the effect of caffeine must be taken into consideration. Additionally, the intake of caffeine from sources other than green tea and the presence or absence of sensitivity to caffeine are issues to be addressed in future research. It is also necessary to consider whether similar effects can be observed in participants with different sleep qualities. Moreover, there are limitations to measuring the physiological phenomenon of sleep in a natural state. This study was conducted in the participants’ homes, respecting their individual lifestyles.

## 5. Conclusions

This study revealed that regular consumption of green tea for 4 weeks had no effect on EEG data. Secondary measurements of subjective sleep quality suggested that it may have a positive effect on mental health. We believe this effect is due to the theanine contained in the leaves of *Camellia sinensis* L., which has been shown to counteract the sleep-disturbing effects of caffeine. However, it is important to be mindful of the amount of theanine consumed. Additionally, this study demonstrated that reducing stress may improve sleep quality. Matcha, which is rich in theanine, may contribute to enhancing mental health. In the future, we will focus on reducing stress in daily life to improve sleep quality and explore the potential benefits of foods, such as green tea, that contain both caffeine and theanine.

## Figures and Tables

**Figure 1 nutrients-16-02907-f001:**
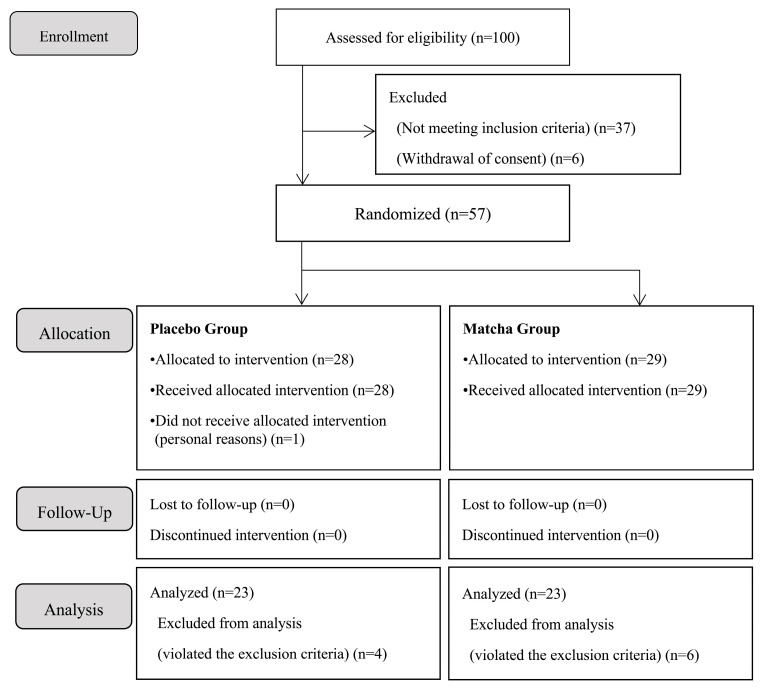
Study flow diagram.

**Table 1 nutrients-16-02907-t001:** Participant clinical characteristics.

Characteristic	Placebo	Matcha
Number of participants	23	23
Sex (Male/Female)	12/11	11/12
Age (years; range 30–64)	49.2 ± 10	50.7 ± 9.5
PSQI-J score (range 2–9)	4.7 ± 1.5	4.5 ± 1.7
Sleep efficiency (%; range 73.1–96.9)	89.7 ± 4.6	89.1 ± 6.2

Values are presented as mean ± SD.

**Table 2 nutrients-16-02907-t002:** Effect of matcha on objective sleep quality.

		0 Weeks	4 Weeks	*p*-Values	*d* Values
Total sleep time (min)	Placebo	373 ± 39	360 ± 58		
	Matcha	378 ± 33	371 ± 43	0.47	0.22
Sleep latency (min)	Placebo	12.1 ± 9.2	12.8 ± 15		
	Matcha	16.1 ± 20	13.5 ± 12	0.86	0.05
Wake after sleep onset (min)	Placebo	31.9 ± 20	32.6 ± 20		
	Matcha	32.8 ± 21	36.1 ± 22	0.58	0.16
Number of awake after sleep onset (times)	Placebo	83.7 ± 33	77.4 ± 35		
	Matcha	95.3 ± 42	95.4 ± 48	0.16	0.43
Sleep efficiency (%TRT)	Placebo	89.7 ± 4.6	88.6 ± 6.2		
	Matcha	89.1 ± 6.2	88.6 ± 5.5	0.97	0.01
Time spent in bed after waking (s)	Placebo	216 ± 221	320 ± 347		
	Matcha	169 ± 201	148 ± 147 *	0.04	−0.65

Values are presented as mean ± SD. * *p* < 0.05 vs. placebo.

**Table 3 nutrients-16-02907-t003:** Effect of matcha on subjective sleep quality.

			0 Weeks	4 Weeks	Amount of Change from Baseline			Change from Baseline (%)		
						*p*-Values	*d* Values		*p*-Values	*d* Values
PSQI-J		Placebo	4.7 ± 1.5	3.9 ± 1.5 ^#^	−0.9 ± 1.8			−13.1 ± 44		
		Matcha	4.5 ± 1.7	4.0 ± 1.7	−0.6 ± 1.6	0.55	0.18	−7.04 ± 40	0.63	0.14
JESS		Placebo	11.0 ± 6.0	8.1 ± 3.6 ^#^	−2.9 ± 5.6			−11.4 ± 44		
		Matcha	10.3 ± 4.9	8.3 ± 4.7 ^#^	−2.0 ± 3.7	0.54	0.18	−16.4 ± 38	0.68	0.12
OSA-MA	Sleepiness on rising	Placebo	13.6 ± 6.5	17.0 ± 4.2 ^##^	3.48 ± 5.6			71.1 ± 150		
		Matcha	14.7 ± 6.0	18.0 ± 5.9 ^#^	3.30 ± 6.7	0.92	0.03	47.7 ± 103	0.54	0.18
	Initiation and maintenance of sleep	Placebo	13.8 ± 3.7	15.7 ± 3.8 ^#^	1.85 ± 3.5			17.8 ± 34		
		Matcha	14.6 ± 4.7	16.2 ± 4.2	1.62 ± 4.8	0.85	0.06	19.5 ± 44	0.89	0.04
	Frequent dreaming	Placebo	19.4 ± 6.6	21.1 ± 6.7	1.77 ± 5.8			15.4 ± 37		
		Matcha	21.1 ± 8.0	19.7 ± 8.0	−1.45 ± 5.7	0.07	0.56	−2.32 ± 27	0.07	0.55
	Refreshing on rising	Placebo	12.6 ± 6.1	17.8 ±4.4 ^##^	5.12 ± 5.9			96.2 ± 150		
		Matcha	14.3 ± 5.4	17.7 ±5.0 ^#^	3.47 ± 6.9	0.39	0.26	48.7 ± 101	0.21	0.37
	Sleep length	Placebo	16.1 ± 4.9	17.1 ±4.6	0.99 ± 4.6			12.8 ± 37		
		Matcha	15.2 ± 5.8	18.6 ±4.5 ^##^	3.42 ± 4.9	0.09	0.51	40.9 ± 81	0.14	0.44
BDI-II		Placebo	7.7 ± 7.2	7.3 ± 7.2	−0.4 ± 5.5			20.9 ± 116		
		Matcha	9.1 ± 7.8	6.7 ± 6.3 ^#^	−2.4 ± 5.5	0.23	0.36	−28.9 ± 47	0.07	0.56

Values are presented as mean ± SD. ^#^ *p* < 0.05, ^##^ *p* < 0.01 vs. 0 Weeks.

## Data Availability

The original contributions presented in the study are included in the article; further inquiries can be directed to the corresponding author.

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
