# Peer review of "Matcha Does Not Affect Electroencephalography during Sleep but May Enhance Mental Well-Being: A Randomized Placebo-Controlled Clinical Trial"

_nutrients, 2024, doi:10.3390/nu16172907_

Round 1
Reviewer 1 Report
Comments and Suggestions for Authors
I would like to commend the authors for their exemplary work. Although the manuscript is well thought-out and written appropriately, I have a few concerns that require addressing prior to consideration for publication.
1. The Introduction is thorough, but verbose. The technical level of detail may be overwhelming to the reader and the overall goal of the manuscript may be lost due to this. Suggest cutting down to ensure concise description of the background and importance within the Introduction section.
2. The Methods are good, but the Flow Diagram numbers do not make sense, specifically in the allocation section of the placebo group and the analysis, which included exclusion from analysis due to violation of exclusion criteria. Traditionally research participants must meet inclusion and exclusion criteria prior to study enrollment. Clarification of these details are key.
3. The Statistical Analysis section is lacking. The authors should perform normality checks prior to analysis to determine how to appropriately handle the data, i.e., parametric versus non-parametric testing or log-transforming non-normally distributed data prior to formal analysis. Given the variability often associated with diet, sleep, and mental well-being studies, a post-hoc power analysis would be helpful to determine the effects that would be expected versus the effects noted in the analysis given the actual sample size. Further, as you excluded participants once the study was complete, identifying if there were differences in the total sample versus sample analyzed versus same excluded would be helpful to justify if findings were meaningful.
4. The Results section needs to start with Participant Demographics at Baseline. Immediately discussing the sleep results does little in clarifying if this is meaningful to the population examined without having to go back to methods to examine inclusion/exclusion criteria.
5. Tables need to present non significant p values and d values for each comparison made. Only presenting significant or approaching significance p values and d values introduces biases in how you represent the results.
6. The results section also needs greater detail written about the findings, otherwise the reader has to examine large tables that can make it difficult to orient and understand what are the key findings that should be highlighted. You want to guide the reader rather than make the reader explore to a given level.
7. The Discussion, similar to the Introduction, is verbose and extremely technical. Making more clear and concise statements would help.
8. The Conclusion is a place to make clear future direction statements. It's an easy out to state that further investigation should continue this work. However, what are the implications of your findings and how do they guide future work? What do you envision are the next steps for this work? Did you perform this to then build out a larger program, or is this it without any consideration for follow-up studies? These are things that could be further explored in the Conclusion section.
Author Response
Thank you very much for reviewing our manuscript. We appreciate the insightful comments of the reviewers. We have provided point-by-point responses to each of the comments below and made the corresponding revisions to the manuscript document. To make our responses easier to locate, they have been highlighted in red.
- The Introduction is thorough, but verbose. The technical level of detail may be overwhelming to the reader and the overall goal of the manuscript may be lost due to this. Suggest cutting down to ensure concise description of the background and importance within the Introduction section.
Response: Thank you for your thoughtful comment.
I provided detailed information because the social background varies from country to country. I have made the corrections, focusing on sleep and cognitive function.
- The Methods are good, but the Flow Diagram numbers do not make sense, specifically in the allocation section of the placebo group and the analysis, which included exclusion from analysis due to violation of exclusion criteria. Traditionally research participants must meet inclusion and exclusion criteria prior to study enrollment. Clarification of these details are key.
Response: Thank you for your important comment.
In this study, the recruitment criteria required participants to have a consistent regular routine. I confirmed the consistency of their routines by measuring EEG before the intervention. However, some participants exhibited clear changes in their routines when EEG measurements were taken after the intervention. For example, one participant who usually turned off the lights at 12 o'clock turned them off at 3 o'clock. Conversely, some participants went to sleep earlier than usual. These deviations violated the recruitment criteria set as a condition for participating in the study. Consequently, these cases were excluded from the analysis.
In this study, measurements were taken at the participants' homes to capture their natural sleeping state. These excluded examples represent the limitations of the study, and I have listed them as such in the limitations section.
- The Statistical Analysis section is lacking. The authors should perform normality checks prior to analysis to determine how to appropriately handle the data, i.e., parametric versus non-parametric testing or log-transforming non-normally distributed data prior to formal analysis. Given the variability often associated with diet, sleep, and mental well-being studies, a post-hoc power analysis would be helpful to determine the effects that would be expected versus the effects noted in the analysis given the actual sample size. Further, as you excluded participants once the study was complete, identifying if there were differences in the total sample versus sample analyzed versus same excluded would be helpful to justify if findings were meaningful.
Response: Thank you for pointing this out.
I added section 2.6 because there was insufficient information on the statistical analysis. Before the analysis, I conducted a normality check. If normality was confirmed, I performed a t-test; if not, I used the Wilcoxon signed-rank test or the Mann-Whitney U test.
- The Results section needs to start with Participant Demographics at Baseline. Immediately discussing the sleep results does little in clarifying if this is meaningful to the population examined without having to go back to methods to examine inclusion/exclusion criteria.
Response: I appreciate your attention to this matter.
Subject background information has been moved to the Results section.
- Tables need to present non significant p values and d values for each comparison made. Only presenting significant or approaching significance p values and d values introduces biases in how you represent the results.
Response: Thank you for bringing this to my attention.
All p-values and d-values have been recorded.
- The results section also needs greater detail written about the findings, otherwise the reader has to examine large tables that can make it difficult to orient and understand what are the key findings that should be highlighted. You want to guide the reader rather than make the reader explore to a given level.
Response: Thank you for your insightful comment.
I have added a note, especially in section 3.2. I also included Table 3 because the results for the comparison with week 0 were not previously shown.
- The Discussion, similar to the Introduction, is verbose and extremely technical. Making more clear and concise statements would help.
Response: Thank you for your perceptive comment.
I have made the necessary corrections.
- The Conclusion is a place to make clear future direction statements. It's an easy out to state that further investigation should continue this work. However, what are the implications of your findings and how do they guide future work? What do you envision are the next steps for this work? Did you perform this to then build out a larger program, or is this it without any consideration for follow-up studies? These are things that could be further explored in the Conclusion section.
Response: Thank you for your meaningful comment.
I have made the necessary corrections.
Reviewer 2 Report
Comments and Suggestions for Authors
1. More specific information about matcha, including include color, particle size, dispersibility, and flowability.
2. From table 2 and table 3, only “Time spent in bed after waking” presented significant differences, in this case, how to reflect the title “enhance mental well-being”?
3. line 90, “Green tea contains both of these compounds”, all kinds of tea contain both of caffeine and theanine.
4. Compared with tea, what do you think is the main factor that matcha does not affect sleep?
5. Matcha has many grades and contains different levels of theanine and caffeine, What grade range is the matcha used in this study?
Author Response
Thank you very much for reviewing our manuscript. We appreciate the insightful comments of the reviewers. We have provided point-by-point responses to each of the comments below and made the corresponding revisions to the manuscript document. To make our responses easier to locate, they have been highlighted in red.
- More specific information about matcha, including include color, particle size, dispersibility, and flowability.
Response: Thank you for your important comment.
There was a lack of information about matcha, so I have added it to the Introduction section.
- From table 2 and table 3, only “Time spent in bed after waking” presented significant differences, in this case, how to reflect the title “enhance mental well-being”?
Response: Thank you for your valuable comment.
This is the point I wanted to emphasize most. I would not have titled the study "enhance mental well-being" based solely on the significant difference in time spent in bed after waking. There was a tendency for improvement in frequent dreaming, sleep length, and BDI-II scores in OSA-MA with p < 0.1. Frequent dreaming is an item related to stress. BDI-II scores were below 10 in both the placebo and matcha groups, indicating that the subjects were not depressed. There was a tendency for the BDI-II score to improve within the normal range. Stress is one of the factors that affect sleep, and the fact that these indicators showed a tendency for improvement is an important sign reflecting mental health. I believe that these tendencies toward improvement contributed to a sense of having slept well, as indicated by sleep length in OSA-MA and time spent in bed after waking. I also believe that a long time spent in bed after waking may indicate a state of stress. Since this time was shortened, I believe that stress was reduced. Because stress was not directly measured in this study, I titled the study "may enhance mental well-being." Sleep testing that takes stress factors into account is a topic for future study.
- line 90, “Green tea contains both of these compounds”, all kinds of tea contain both of caffeine and theanine.
Response: Your meaningful comment is duly noted.
As you pointed out, all types of tea contain both caffeine and theanine. I have changed "green tea" to "tea tree" to avoid any confusion.
- Compared with tea, what do you think is the main factor that matcha does not affect sleep?
Response: I appreciate your valuable question.
I believe that the amount of theanine is the key factor. Tea contains caffeine, which can induce wakefulness after sleep onset. However, if tea contains a sufficient amount of theanine, this effect can be suppressed. I believe that matcha, with its high theanine content, does not affect sleep in the same way as regular green tea. In addition, the anti-stress effect of theanine, as suggested in this study, may improve mental health and enhance sleep quality, although further research is needed to confirm this.
- Matcha has many grades and contains different levels of theanine and caffeine, What grade range is the matcha used in this study?
Response: Thank you for your insightful question.
Matcha itself is recognized as a high-grade type of green tea. However, there are currently no standardized grades for matcha. Some may categorize matcha based on factors such as price or amino acid content, but there is no consensus on grading. If a grade is displayed on matcha, it is often very vague and may be specific to Japan. Therefore, in this study, I have listed the amounts of theanine, catechin, and caffeine but not a grade.
Reviewer 3 Report
Comments and Suggestions for Authors
The authors present interesting research in which they aimed to evaluate the effects of Matcha on sleep quality and mental well-being.
Although they have done a good job in substantiating the manuscript and presenting their research, I still have some methodological gaps that I would like to have explained.
I would especially like to see the recommendations of the STROBE guideline followed and to include which items were taken into account and where they can be found in the manuscript.
What measures were taken to reduce possible biases? Was the distribution followed by the data of the study variables studied?
The study variables should be better defined. Surely the authors know their design and study variables very well, but they are not very clear to the readers. I would like them to be more clearly stated and defined, as well as providing data on the reliability of the tests in the study population.
With those problems solved, I think this manuscript could have potential for acceptance.
Best Regards
Author Response
Thank you very much for reviewing our manuscript. We appreciate the insightful comments of the reviewers. We have provided point-by-point responses to each of the comments below and made the corresponding revisions to the manuscript document. To make our responses easier to locate, they have been highlighted in red.
The authors present interesting research in which they aimed to evaluate the effects of Matcha on sleep quality and mental well-being.
Although they have done a good job in substantiating the manuscript and presenting their research, I still have some methodological gaps that I would like to have explained.
I would especially like to see the recommendations of the STROBE guideline followed and to include which items were taken into account and where they can be found in the manuscript.
What measures were taken to reduce possible biases? Was the distribution followed by the data of the study variables studied?
The study variables should be better defined. Surely the authors know their design and study variables very well, but they are not very clear to the readers. I would like them to be more clearly stated and defined, as well as providing data on the reliability of the tests in the study population.
With those problems solved, I think this manuscript could have potential for acceptance.
Response: I appreciate your helpful feedback.
As you pointed out, the study design was difficult to understand, so I have included it in the title. This study is an RCT, as described in section 2.4, “Study design.” Therefore, I have structured it in accordance with the CONSORT statement. In section 2.3, “Participants,” I have clearly stated the inclusion and exclusion criteria, and in Figure 1, I have provided the study flow diagram to ensure the transparency of the study.
Round 2
Reviewer 1 Report
Comments and Suggestions for Authors
I appreciate the author's time and effort in making requested revisions. Excellent work and I look forward to the published product!
Reviewer 2 Report
Comments and Suggestions for Authors
The revision is acceptable, I do not have any other questions.
Reviewer 3 Report
Comments and Suggestions for Authors
The authors made the necessary changes and/or made the appropriate justifications. I believe that the manuscript could be accepted if the other reviewers and the editor consider it so.